# Associations of the Neutrophil-to-Lymphocyte Ratio and Platelet-to-Lymphocyte Ratio with Osteoporosis: A Meta-Analysis

**DOI:** 10.3390/diagnostics12122968

**Published:** 2022-11-28

**Authors:** Yu-Cheng Liu, Tzu-I Yang, Shu-Wei Huang, Yi-Jie Kuo, Yu-Ping Chen

**Affiliations:** 1College of Medicine, Taipei Medical University, Taipei 110301, Taiwan; 2Department of General Medicine, Changhua Christian Hospital, Changhua 500209, Taiwan; 3Department of Orthopedics, Wan Fang Hospital, Taipei Medical University, Taipei 110301, Taiwan; 4Department of Orthopedics, School of Medicine, College of Medicine, Taipei Medical University, Taipei 110301, Taiwan

**Keywords:** neutrophil-to-lymphocyte ratio, platelet-to-lymphocyte ratio, osteoporosis, bone mineral density

## Abstract

Osteoporosis is characterized by low bone mass and increased bone fragility. Numerous studies have suggested that inflammation contributes to its pathogenesis. The neutrophil-to-lymphocyte ratio (NLR) and platelet-to-lymphocyte ratio (PLR) are simple, noninvasive biomarkers that can reflect the inflammation status on human body. However, evidence on their associations with osteoporosis remains scant. The PubMed, Embase, and Cochrane Library databases were searched for relevant studies from their inception to April 2022. Observational studies providing complete NLR or PLR data in both the osteoporosis and normal bone mineral density (BMD) groups were included. Studies involving individuals at risk of secondary osteoporosis or restricted to a certain disease population were excluded. The main outcome was the associations of NLR and PLR with osteoporosis. Between-group differences were measured using mean differences (MDs) and 95% confidence intervals (CIs). In our analysis, both NLR and PLR were significantly higher in the osteoporosis group (MD = 0.494, 95% CI: 0.339–0.649, *p* < 0.0001; MD = 23.33, 95% CI: 4.809–41.850, *p* = 0.014, respectively) than in the normal BMD group. NLR was significantly higher in postmenopausal women with osteoporosis (MD = 0.432, 95% CI: 0.309–0.544, *p* < 0.0001). Our findings suggest the associations of NLR and PLR with osteoporosis. NLR and PLR constitute potential targets in osteoporosis screening.

## 1. Introduction

Osteoporosis is a metabolic skeletal disease characterized by reduced bone mass, increased bone fragility, and the microarchitectural deterioration of bone tissue, which result in susceptibility to fracture [1]. Osteoporosis is especially common in older men and postmenopausal women; an overview revealed that one-third of women aged more than 50 years will experience osteoporotic fractures at some point in their lifetime, as will one-fifth of older men [2]. Osteoporotic fractures account for 0.83% of the global burden of noncommunicable diseases, with an osteoporotic fracture occurring every 3 seconds [3]. Considering the rapid aging of the population globally, osteoporosis constitutes a considerable public health concern because of its associated fracture risks and care demands [4].

Osteoporosis is a multifactorial disease, which may be attributed to endocrine, metabolic, and mechanical factors. Risk factors for osteoporosis include older age, being female, estrogen deficiency, and long-term use of medications such as glucocorticoids [5]. Emerging evidence suggests that inflammation plays an essential role in bone turnover [6]. Chronic inflammation and aging-related immune system remodeling are potentially associated with the pathogenesis of osteoporosis [7]. Experimental studies have indicated that several proinflammatory cytokines, including interleukin (IL)-1, IL-6, and tumor necrosis factor (TNF), are linked to reduced bone mass and increased fracture risk [6]. High-sensitivity C-reactive protein (CRP), a commonly used clinical marker of systemic inflammation, is correlated with bone mineral density (BMD) and osteoporotic fracture in several immune and inflammatory diseases as well as in healthy individuals, which suggests a relationship between subclinical systemic inflammation and osteoporosis [8]. Despite the strong association between inflammatory markers and bone loss, researchers have not determined which marker is the most critical for bone health and the most suitable for use in clinical osteoporosis screening [9].

The neutrophil-to-lymphocyte ratio (NLR) and platelet-to-lymphocyte ratio (PLR), the ratio of absolute neutrophil and platelet count to absolute lymphocyte count, are simple, cost-effective, and noninvasive biomarkers that can reflect the inflammation status on human body [10]. Several studies have shown that they are useful markers in the assessment of inflammatory response and disease activity in autoimmune disorders such as systemic lupus erythematosus and rheumatoid arthritis [11]. However, few studies have reported on the correlations of NLR and PLR with osteoporosis even with increasing evidence of the role of inflammation in the pathogenesis of osteoporosis. No conclusive evidence has been provided on the associations of these two biomarkers with osteoporosis.

Considering the limited and inconclusive state of the literature, we performed a meta-analysis probing the differences in NLR and PLR between individuals with and without osteoporosis. We hypothesized that both NLR and PLR were higher in individuals with osteoporosis.

## 2. Materials and Methods

### 2.1. Search Strategy and Identification of Eligible Studies

This study was performed in adherence to the Preferred Reporting Items for Systematic Reviews and Meta-analyses statement (PRISMA) [12]. The two reviewers (Y.-C.L. and T.-I.Y.) searched the PubMed, Embase, and Cochrane Library databases for relevant publications with no language restrictions applied. The following search keywords were employed: (“neutrophil to lymphocyte ratio” or “neutrophil lymphocyte ratio” or “NLR” or “platelet to lymphocyte ratio” or “platelet lymphocyte ratio” or “PLR”) and (“osteoporosis” or “bone mineral density” or “bone loss”). All databases were searched from their inception to April 2022. To broaden the search, the reference lists of the studies relevant to this topic were screened; we also identified additional studies by manually searching in Google Scholar. This study had been registered in the International Prospective Register of Systematic Reviews (registration no. CRD42022330409).

### 2.2. Inclusion and Exclusion Criteria

The inclusion criteria were as follows: (1) published observational studies with full texts, (2) studies measuring BMD with validated methods such as dual energy X-ray absorptiometry and providing clear definition of osteoporosis (such as a T score of ≤−2.5 standard deviations), and (3) studies providing complete data on NLR or PLR in both the osteoporosis and normal BMD groups.

The exclusion criteria were as follows: (1) studies including patients at risk of secondary osteoporosis (such as those with parathyroid diseases), (2) studies including patients with certain clinical conditions which may interfere with NLR or PLR values (such as cancer, acute infectious diseases, or hematological disorders), (3) studies including patients restricted to a certain disease population, or (4) studies in which NLR or PLR data corresponding to the osteoporosis and normal BMD group were unavailable or incomplete.

### 2.3. Data Extraction and Appraisal of Methodological Quality

The two reviewers (Y.-C.L. and T.-I.Y.) independently identified potentially relevant studies, reviewed the full texts of the articles, and extracted baseline and outcome information from either the datasets or figures and tables. Extracted information included the author, country, published year, study design, sample size, sex distribution, mean age, methodology for the measurement of bone mineral density, prediction model for osteoporosis (NLR or PLR), and complete NLR or PLR data. For studies reporting NLR and PLR data as median with full range, a conversion into mean with standard deviation was conducted following a validated statistical method [13].

The methodological quality of the included studies was assessed using a modified version of the Newcastle–Ottawa scale (NOS) [14], adapted from NOS for cohort studies to provide quality assessment of cross-sectional studies. The modified NOS comprises three domains, namely selection, comparability, and outcome, with a total maximum score of ten. High-quality studies had a total score of seven to ten; studies that scored below four were regarded as being low quality. The two reviewers (Y.-C.L. and T.-I.Y.) independently evaluated the methodological quality of the included studies. Any disagreements in the data extraction and appraisal process were resolved through discussion with the third reviewer (Y.-P.C.).

### 2.4. Outcomes

The main outcome of interest was the associations of NLR and PLR with osteoporosis, evaluated by differences in NLR and PLR values between the osteoporosis and normal BMD groups. We performed subgroup analysis to furtherly determine the association of NLR with osteoporosis in postmenopausal population. Considering that age and diabetes mellitus (DM) are confounders of NLR value [15,16], which may affect our results, we conducted meta regression to evaluate their effects on the between-group differences in NLR.

### 2.5. Statistical Analysis and Data Synthesis

For meta-analysis, the data were analyzed using Comprehensive Meta-Analysis Software (Biostat, Englewood, NJ, USA). Differences in NLR and PLR between the osteoporosis and normal BMD groups were estimated using mean differences (MDs) and 95% confidence intervals (CIs); a *p* value of < 0.05 indicated significance. The random-effect model was selected for analysis. Heterogeneity among the included studies was examined using the standard chi-square test and *I*^2^ test; significance was set at *p* < 0.05 for standard chi-square test. Publication bias was evaluated using a funnel plot and through Egger’s test.

## 3. Results

Figure 1 presents the flowchart of study screening and selection. Our initial search of the PubMed, Embase, and Cochrane Library databases, as well as of Google Scholar, yielded 473 studies. After the removal of duplicates, 410 studies were screened according to their titles and abstracts, after which 386 studies were excluded. For the remaining 24 studies, full text evaluation was performed. Of these studies, 14 were excluded for the following reasons: full texts were not available for three studies, two were non-related studies investigating topics out of interest, eight studies did not provide data on NLR or PLR, and one study provided significant outlying NLR value, which was significantly higher for our target population. The remaining ten studies were included in the meta-analysis [17,18,19,20,21,22,23,24,25,26]. All the included studies were published in English.

Table 1 presents details on the baseline characteristics of the included studies. The ten studies were all cross-sectional design, published between 2009 and 2021, conducted in China, Korea, Oman, and Turkey, and comprised 2616 individuals (1830 with osteoporosis, 786 with normal BMD) in total. Six studies comprised only postmenopausal women [17,18,20,21,25,26], and the other four studies included adults with no restrictions on sex [19,22,23,24]. Most studies measured bone mineral density by using dual energy X-ray absorptiometry, and recorded as T-score; measured sites included lumbar spine, pelvis, femoral neck, proximal femur, and total femur. All studies provided data on NLR in both the osteoporosis and normal BMD groups; five studies provided data on PLR [17,18,19,21,22].

The methodological quality of the included studies was assessed using the modified NOS, with results shown in Appendix A. Of the ten studies, eight were determined to be high quality [18,19,21,22,23,24,25,26]; the remaining two studies were regarded as moderate quality due to poor baseline control for potential confounders of NLR and PLR [17,20]. 

### 3.1. Association of NLR with Osteoporosis

#### 3.1.1. Mean Difference in NLR between the Osteoporosis and Normal BMD Groups

As mentioned, all included studies provided data on NLR in both the osteoporosis and normal BMD groups. Our analysis showed that NLR was significantly higher in the osteoporosis group than in the normal BMD group (MD = 0.494, 95% CI: 0.339–0.649, *p*  <  0.0001; Figure 2). The heterogeneity among the included studies was nonsignificant (*I*^2^: 46%, *p* = 0.055; Figure 2).

#### 3.1.2. Subgroup Analysis

In subgroup analysis, we furtherly analyzed the difference in NLR between the osteoporosis and normal BMD groups in postmenopausal women. Six of the included studies provided NLR data in both groups and involved postmenopausal women only [17,18,20,21,25,26]. The pooled results showed that NLR was significantly higher in the osteoporosis group (MD = 0.432, 95% CI: 0.309–0.554, *p* <  0.0001; Figure 3), with low heterogeneity detected across the included studies (*I*^2^: 0%, *p* = 0.497; Figure 3).

#### 3.1.3. Meta Regression

We performed meta regression to evaluate the potential confounding effects of age and the inclusion of DM patients on the between-group differences in NLR. The mean age of the enrolled individuals in the included studies ranged from 54.2 to 74.9 years. Four of the studies excluded individuals with DM [21,22,24,25], whereas others did not [17,18,19,20,23,26]. No significant associations were identified through meta regression (regression coefficient = −0.0083, *p* = 0.478; regression coefficient = −0.1417, *p* = 0.444, respectively; Table 2). Scatterplots were shown in the Appendix A.

### 3.2. Association of PLR with Osteoporosis

#### Mean Difference in PLR between the Osteoporosis and Normal BMD Groups

Five of the included studies provided PLR data in both the osteoporosis and normal BMD groups [17,18,19,21,22]. PLR was significantly higher in the osteoporosis group than in the normal BMD group (MD = 23.33, 95% CI: 4.809–41.850, *p* = 0.014; Figure 4). Significant heterogeneity was detected across the analyzed studies (*I*^2^: 79%, *p* =  0.001; Figure 4).

### 3.3. Publication Bias

We evaluated publication bias in analyses involving five or more studies. Funnel plots of all the examined analyses were shown in Appendix A. No publication biases were detected through Egger’s test.

## 4. Discussion

In the present study, our analysis showed that both NLR and PLR were higher in individuals with osteoporosis compared to those with normal BMD. In the subgroup analysis of postmenopausal population, NLR was significantly higher in postmenopausal women with osteoporosis. Through meta regression, we found that the differences in NLR between the osteoporosis and normal BMD groups would not be affected by age and the inclusion of patients with DM. To the best of our knowledge, this is currently the first and the most extensive study investigating the associations of NLR and PLR with osteoporosis. Our findings support our hypothesis, demonstrating the link between these two inflammatory biomarkers and osteoporosis.

The concept that inflammation may contribute to the pathogenesis of osteoporosis has been proposed in recent years [7,27]. The two dominant causes of primary osteoporosis, namely estrogen deficiency and aging, share the same pathological pathway as inflammation in the progression of BMD loss [28,29]. Both estrogen deficiency and aging are associated with immune dysregulation, characterized by increased circulating level of numerous proinflammatory cytokines, including IL-1, IL-6, and TNF, which may result in a chronic state of systemic and subclinical inflammation [28,30]. With such inflammatory stimuli, blood cells including neutrophils and platelets may be activated and recruited [31,32], which can be reflected by higher NLR and PLR values; furthermore, the increased circulating levels of proinflammatory cytokines and immune cells may all directly or indirectly affect osteoclastogenesis through various pathways, and thereby the bone resorption [28,33]. In brief, higher level of inflammation associated with estrogen deficiency and aging in some individuals, regardless of an individual’s general health condition or other comorbidities, may be associated with a relatively overactive immune response, reflected by higher NLR and PLR, and lead to the dysregulation of bone homeostasis and eventual osteoporosis.

Another possible interpretation can be given for the higher NLR and PLR of individuals with osteoporosis; that is, osteoporosis may result in higher NLR and PLR values because of its association with impaired hematopoiesis and an unbalanced decline in myeloid and lymphoid cells [29,34]. Two in vivo studies have reported an increase in hematopoietic stem cells (HSCs) in bone marrow with increased osteoblast populations in genetic mutant mice; on the contrary, the targeted deletion of osteoblasts may lead to the subsequent loss of HSCs [35,36]. In another study conducted in mice, with the depletion of osteoblasts, both myeloid and lymphoid cells decreased, but the reduction was gradual in myeloid cells and sharp in lymphoid cells [37]. Taken together, the results suggest that the functional decline of osteoblasts with the progression of osteoporosis may be associated with impaired hematopoiesis in the human bone marrow. This impaired state is characterized by the unbalanced reduction of myeloid and lymphoid cells, which may result in higher NLR and PLR values.

In our study, we investigated the difference in NLR between individuals with and without osteoporosis in postmenopausal population; NLR was significantly higher in postmenopausal women with osteoporosis. In addition, we conducted meta regression to examine whether age and the inclusion of patients with DM would affect the differences in NLR between the osteoporosis and normal BMD groups. Both age and DM have been reported to be confounding factors of NLR value [15,16]; older adults and individuals with DM exhibit higher NLR value. Based on our results, no significant associations were found between these two confounders and the between-group differences in NLR. The result regarding DM is supported by a previous study, in which NLR was significantly higher in patients with osteoporosis in DM population [38]. In sum, our findings support the potential use of NLR in osteoporosis screening regardless of age or the presence of DM, and particularly in postmenopausal population.

In our analyses of NLR, heterogeneity among the analyzed studies was generally nonsignificant, suggesting the strong association between NLR and osteoporosis. However, significant heterogeneity was detected in our analysis of PLR. Due to the small number of analyzed studies, we failed to find possible explanations for the high heterogeneity through subgroup analysis or other approaches. In our opinion, the heterogeneity may be attributed to the between-study variations, such as inclusion/ exclusion criteria and enrolled patients’ baseline characteristics or comorbidities. Furthermore, we performed mathematical conversion to obtain adequate PLR data for outcome pooling for some studies, which may also affect the results, and thereby the heterogeneity.

Some limitations of our study need to be addressed. Firstly, as mentioned, the number of the included studies investigating the association of PLR with osteoporosis is small, which makes our findings regarding PLR less conclusive and comprehensive. Secondly, although we have provided possible explanations for the increase of NLR and PLR in patients with osteoporosis, no causal relationships between them can be established because of the cross-sectional design of all included studies. Thirdly, due to insufficient data, we could not identify cutoff values of NLR and PLR with acceptable sensitivity and specificity for predicting osteoporosis; considering the easy accessibility of these two biomarkers, this may be a future research direction since clinicians may start early interventions, such as lifestyle modifications, for those at higher risk of osteoporosis, based on their NLR or PLR values.

## 5. Conclusions

Both NLR and PLR are higher in individuals with osteoporosis in comparison to those with normal BMD, which suggests the associations of these two inflammatory biomarkers with osteoporosis. Further research is warranted to elucidate the roles of NLR and PLR as predictors of osteoporosis, as well as to identify their applicability to osteoporosis screening in clinical practice.

## Figures and Tables

**Figure 1 diagnostics-12-02968-f001:**
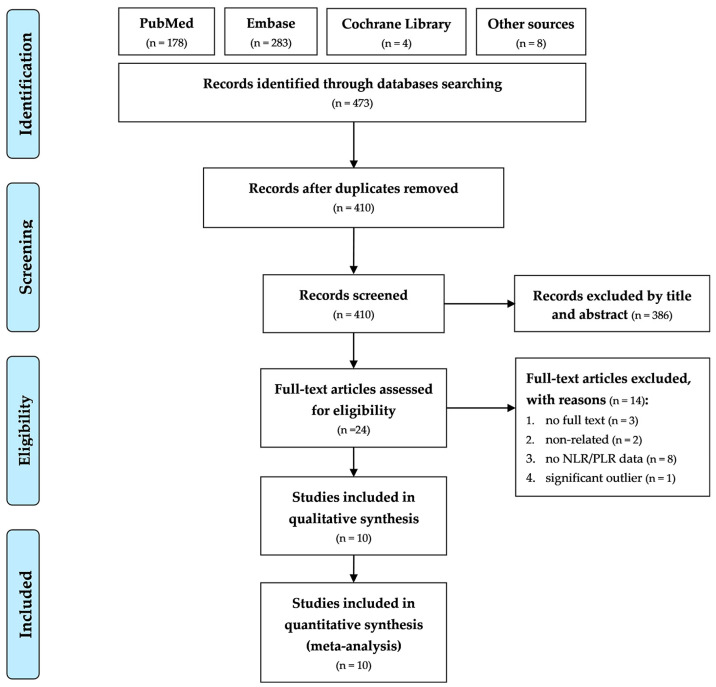
PRISMA flowchart of the selection of included studies.

**Figure 2 diagnostics-12-02968-f002:**
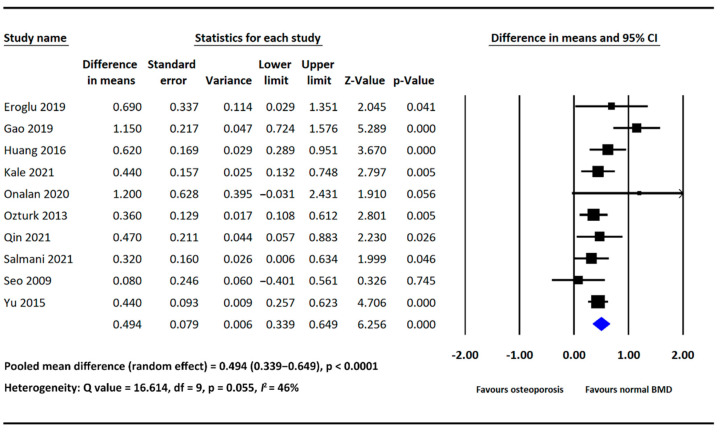
Forest plot of the mean difference in NLR between the osteoporosis and normal BMD groups [17,18,19,20,21,22,23,24,25,26].

**Figure 3 diagnostics-12-02968-f003:**
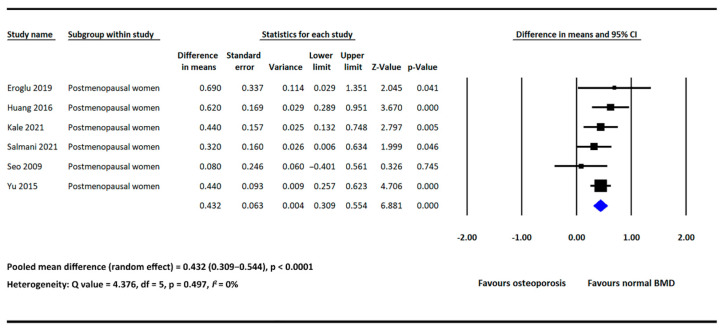
Forest plot of the mean difference in NLR between the osteoporosis and normal BMD groups in postmenopausal women [17,18,20,21,25,26].

**Figure 4 diagnostics-12-02968-f004:**
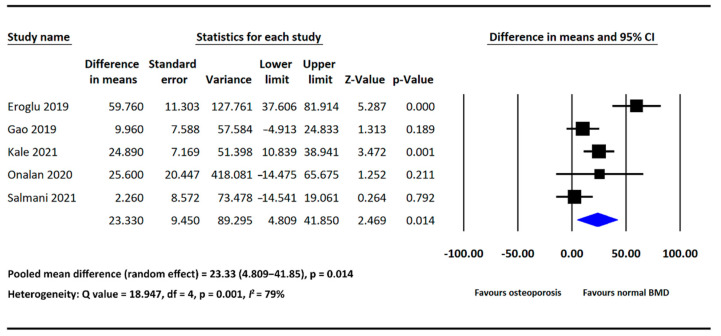
Forest plot of the mean difference in PLR between the osteoporosis and normal BMD groups [17,18,19,21,22].

**Table 1 diagnostics-12-02968-t001:** Baseline characteristics of the included studies.

Author (Year)	Country	Study Design	Sample (N)	Female (*n*, %)	Mean Age (Years)	Methodology for BMD Measurement; Site	Model
Eroglu et al., (2019) [18]	Turkey	cross-sectional	140	140, 100%	54.21	DEXA; lumbar spine, total femur, femoral neck	NLR, PLR
Gao et al., (2019) [19]	China	cross-sectional	210	119, 56.7%	57.16	DEXA; lumbar spine, femoral neck	NLR, PLR
Huang et al., (2016) [20]	China	cross-sectional	173	173, 100%	62.2	DEXA; lumbar spine, left total femur	NLR
Kale et al., (2021) [21]	Turkey	cross-sectional	74	74, 100%	59.62	DEXA; lumbar spine, femoral neck	NLR, PLR
Onalan et al., (2020) [22]	Turkey	cross-sectional	215	177, 82.3%	73.73	DEXA; not declared	NLR, PLR
Ozturk et al., (2013) [23]	Turkey	cross-sectional	1011	608, 60.1%	72.63	DEXA; lumbar spine, total femur, femoral neck	NLR
Qin et al., (2021) [24]	China	cross-sectional	29	20, 69.0%	59.96	QCT; not declared	NLR
Salmani et al., (2021) [17]	Oman	cross-sectional	286	286, 100%	64.41	DEXA; lumbar spine, pelvis, femoral neck	NLR, PLR
Seo et al., (2009) [25]	Korea	cross-sectional	66	66, 100%	56.53	DEXA; lumbar spine, proximal femur	NLR
Yu et al., (2015) [26]	China	cross-sectional	412	412, 100%	74.95	DEXA; lumbar spine, femoral neck	NLR

Abbreviations: BMD, bone mineral density; DEXA, dual energy X-ray absorptiometry; NLR, neutrophil-to-lymphocyte ratio; PLR, platelet-to-lymphocyte ratio; QCT, quantitative computed tomography.

**Table 2 diagnostics-12-02968-t002:** Meta regression table.

Moderators	Regression Coefficient	95% CI, Lower Limit	95% CI, Upper Limit	*p* Value
Age	−0.0083	−0.0312	0.0146	0.478
DM *	−0.1417	−0.5044	0.2209	0.444

Abbreviations: CI, confidence interval; DM, diabetes mellitus. * Excluding individuals with DM or not.

## Data Availability

All data generated or analyzed in this study are included in the published article and its Appendix A.

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
