# Peer review of "Associations of the Neutrophil-to-Lymphocyte Ratio and Platelet-to-Lymphocyte Ratio with Osteoporosis: A Meta-Analysis"

_diagnostics, 2022, doi:10.3390/diagnostics12122968_

Round 1

Reviewer 1 Report

This meta-analysis focuses on the association between two markers of inflammation (neutrophil-to-lymphocyte ratio and platelet-to-lymphocyte ratio) and osteoporosis. In total, 10 studies were included in their analysis. The analysis appears to be scientifically sound. However, some more information is warranted regarding some of the exclusions. Overall, the conclusions appear to be justified by the results. English language is excellent.

Minor comments:

1. Results, ln 137-138: Please clarify what is meant by “irrelevant”.

2. Results, ln 139: Please provide more information about this outlier and in what way it differed in order to justify its exclusion.

Grammar:

1. Introduction, ln 49-50: Because tumor necrosis factor-beta is now called lymphotoxin-alpha, tumor necrosis factor-alpha can be called simply “tumor necrosis factor”. Source: Wang X, Yang C, Körner H, Ge C. Tumor Necrosis Factor: What Is in a Name? Cancers (Basel). 2022 Oct 27;14(21):5270. doi: 10.3390/cancers14215270. PMID: 36358688; PMCID: PMC9656125.

2. Suggest replacing “NLR/ PLR” with “NLR or PLR” or “NLR and PLR” depending on the context throughout. The use of “NLR/ PLR” may be confused as a ratio between these two metrics.

Author Response

Please see the attachment: Response to Reviewer 1 Comments-diagnostics 2047821.

Reviewer 2 Report

Very Respected Authors,

After carefully reading your manuscript I determined that the whole manuscript is well structured, the section Introduction is well written.  The objective is clear and the section Material and Methods is well designed. The findings are well presented in the section Results.  Conclusion is in agreement with the objective and the findings.  I suggest to find some more new references not older than 3 years.

Author Response

Please see the attachment: Response to Reviewer 2 Comments-diagnostics 2047821
